# Anti-Inflammatory and Antioxidant Effects of Irigenen Alleviate Osteoarthritis Progression through Nrf2/HO-1 Pathway

**DOI:** 10.3390/ph17101268

**Published:** 2024-09-26

**Authors:** Xuan Fang, Hongqi Zhao, Tao Xu, Hua Wu, Gaohong Sheng

**Affiliations:** Department of Orthopedics, Tongji Hospital, Tongji Medical College, Huazhong University of Science and Technology, Wuhan 430074, China; m202276403@hust.edu.cn (X.F.); m202176249@hust.edu.cn (H.Z.); xt2023@hust.edu.cn (T.X.); wuhua@hust.edu.cn (H.W.)

**Keywords:** irigenin, osteoarthritis, inflammation, oxidant stress, Nrf2

## Abstract

Background/Objectives: Osteoarthritis (OA) is a prevalent degenerative disease globally, characterized by cartilage degradation and joint dysfunction. Current treatments are insufficient for halting OA progression. Irigenin (IRI), a flavonoid extracted from natural plants with anti-inflammatory and antioxidant properties, has demonstrated potential in mitigating inflammation and oxidative stress in various diseases; however, its effects on OA remain unexplored. This study aims to evaluate the therapeutic effects of IRI on OA through in vivo and in vitro experiments and to elucidate the underlying molecular mechanisms. Methods: In vitro, chondrocytes were exposed to hydrogen peroxide (H_2_O_2_) to induce an oxidative stress environment and were then treated with IRI. Western blotting, RT-qPCR, immunofluorescence staining assays, flow cytometry, and apoptosis assays were employed to assess the effects of IRI on chondrocyte matrix homeostasis, inflammatory response, and apoptosis. In vivo, an OA rat model was treated with regular IRI injections, and therapeutic effects were evaluated using micro-CT, histological staining, and immunohistochemistry assays. Results: IRI treatment restored matrix homeostasis in chondrocytes and effectively suppressed H_2_O_2_-induced inflammation and apoptosis. Subsequent studies further revealed that IRI exerts its therapeutic effects by activating the Nrf2/HO-1 pathway. Inhibition of Nrf2 expression in chondrocytes partially blocked the anti-inflammatory and antioxidant effects of IRI. In the OA rat model, regular IRI injections effectively ameliorated cartilage degeneration. Conclusions: This study identifies IRI as a promising strategy for OA treatment by modulating inflammation and apoptosis through the Nrf2/HO-1 pathway.

## 1. Introduction

Osteoarthritis (OA) is the most prevalent joint disorder, affecting approximately 500 million people globally [1,2]. The breakdown of cartilage, which involves the imbalance of anabolic activities and catabolic abilities in chondrocytes, is a major characteristic of OA and induces impairment of joint function [2,3]. In addition, low-grade inflammation contributes to OA pathogenesis and exacerbates arthritis pain [4,5]. Despite the pain and mobility issues associated with OA, current treatments are inadequate for halting disease progression [6]. Therefore, there is an urgent need to identify effective therapies to prevent cartilage degeneration [7].

Irigenin (IRI) is a flavonoid extracted from various natural plants, including *Belamcanda chinensis* [8], Crocus leaf [9], and Ukrainian Iris [10]. The anti-inflammatory properties of IRI have been demonstrated, and it has shown promise in the attenuation of diseases related to inflammation. For example, Guo et al. reported that IRI effectively suppressed the production of inflammatory cytokines in cardiomyocytes and mitigated doxorubicin-induced cardiac injury [11]. Similarly, Liu et al. identified that IRI reduced the levels of TNF-α, IL-1β, IL-6, and IL-18 in bronchoalveolar lavage fluid and also alleviated lipopolysaccharide-induced acute lung injury [12]. In addition, IRI exhibited antioxidant activities by decreasing intracellular reactive oxygen species (ROS) levels [13]. The reduced ROS levels may be attributed to the capacity of IRI to enhance the activities of key antioxidant enzymes, including glutathione peroxidase, catalase, and superoxide dismutase [14]. Further studies have indicated that IRI reduced inflammatory cytokines and intracellular ROS levels in a nuclear factor erythroid 2-related factor 2 (Nrf2)-dependent manner [13,14]. Although IRI has demonstrated curative effects on various diseases, studies about its effect on inhibiting OA progression are currently absent.

Nrf2 is a well-established transcription factor that protects against oxidative stress and inflammation by regulating the expression of genes involved in cellular redox balance [15,16]. Nrf2 plays a critical role in cellular defense against oxidative stress and becomes a potential therapeutic target in various diseases including neurodegenerative diseases [17], inflammatory bowel disease [18,19] and cardiac dysfunction [20]. Nrf2 also maintains articular cartilage homeostasis and prevents cartilage degeneration by regulating redox balance in chondrocytes [21,22,23]. Hence, Nrf2-activating pharmaceutical agents have the potential to alleviate OA progression. Dong et al. identified that Nrf2 activation by the synthetic triterpenoid CDDO-Im effectively inhibited cartilage destruction and reduced serum levels of inflammatory cytokines [24]. In addition, activation of Nrf2 by natural compounds could reduce chondrocyte apoptosis induced by oxidative stress [25,26]. Thus, it is reasonable to hypothesize that IRI may hold promise for ameliorating OA progression, given its role in resolving inflammation through Nrf2 regulation.

In this study, we identified IRI, which exhibits anti-inflammatory and antioxidant activities, as having the potential to alleviate the progression of osteoarthritis (OA) by inhibiting chondrocyte catabolism and apoptosis. Further, it was verified that the therapeutic effects of IRI on OA chondrocytes were partially dependent on the regulation of Nrf2. Therefore, IRI provides a promising option for the management of OA.

## 2. Results

### 2.1. Effects of IRI on Chondrocyte Viability and Selection of the Optimal Treatment Concentration of IRI

The structure of IRI is presented in Figure 1A. A CCK-8 kit and Calcein/PI staining assay were employed to evaluate the influences of IRI on chondrocyte viability. The results indicated that the viability of chondrocytes was significantly inhibited by 80 μM of IRI (Figure 1B), whereas IRI treatment at concentrations of 5 μM, 10 μM, 20 μM, and 40 μM did not notably affect cell viability for 24 h (Figure 1B). Moreover, the results of Live/Dead staining also confirmed that concentrations of IRI of 40 μM and below did not significantly induce cell death (Figure 1C). Then, Western blotting was employed to investigate the optimal treatment concentration of IRI. The results showed that 40 μM of IRI exerted the most pronounced effect on inhibiting inflammation and catabolism (Figure 1D,E). Therefore, 40 μM of IRI was selected for further experimentation.

### 2.2. IRI Inhibited the Inflammation Response and Restored Matrix Homeostasis in the H_2_O_2_-Stimulated Chondrocytes

To further explore the therapeutic effects of IRI on extracellular matrix (ECM) metabolism and inflammation response during OA progression, we exposed chondrocytes to 400 µM of H_2_O_2_ to induce an oxidative-stress-mediated chondrocyte degradation phenotype [27]. Cartilage-specific markers were assessed using Western blotting, RT-qPCR, and immunofluorescence (IF) staining. After stimulation with H_2_O_2_ for 24 h, inflammatory markers, such as COX2 and iNOS, as well as catabolic marker MMP13, were elevated at the protein level, while the anabolic marker COL2 was reduced (Figure 2A). However, the cartilage degradation and inflammatory phenotype were reversed following treatment with 40 μM of IRI (Figure 2A). In addition, the RT-qPCR analysis showed that the IRI decreased the mRNA levels of *Mmp3*, *Mmp13*, *Nos2* and *Cox2*, which were upregulated following H_2_O_2_ exposure, and increased the expression of *Acan* and *Col2* (Figure 2B–G). Similar results were also observed in IF staining of MMP13, iNOS, and COL2 (Figure 2H,I).

### 2.3. IRI Attenuated Apoptosis and Promoted Proliferation in H_2_O_2_-Induced Chondrocytes

Chondrocyte apoptosis plays a key role in OA progression [28]. Previous experiments have shown that IRI exhibits significant anti-apoptotic effects in various diseases [11,14,29]. Nevertheless, the specific anti-apoptotic mechanisms of IRI in OA chondrocytes are not fully understood. To investigate this, chondrocytes were exposed to oxidative stress induced by H_2_O_2_ for 24 h to induce cell apoptosis and further explore the therapeutic mechanisms of IRI on OA. As shown in Figure 3A,B, IRI treatment effectively reversed the H_2_O_2_-induced increase in cleaved caspase-3 and BAX while upregulating the protein level of BCL2. Flow cytometric analysis revealed that H_2_O_2_ stimulation increased the proportion of apoptotic cells, while IRI treatment significantly suppressed the rate of apoptotic cell production (Figure 3C,D). Furthermore, TUNEL-positive cells were increased following H_2_O_2_ stimulation and reversed by IRI (Figure 3E,F). In addition, proliferation inhibition of chondrocytes following H_2_O_2_ exposure was mitigated by IRI treatment (Figure 3G,H). Live/Dead staining further demonstrated that IRI treatment significantly decreased the amount of H_2_O_2_-induced dead cells (Figure 3I,J).

### 2.4. IRI Decreased Oxidative Stress through Activating the Nrf2/HO-1 Axis in Chondrocytes

Previous research has demonstrated that IRI exerts neuroprotective effects through the activation of the Keap1/Nrf2 pathway [14]. Therefore, we hypothesized that IRI alleviated H_2_O_2_-induced oxidative stress damage in chondrocytes by modulating Nrf2. A molecular docking analysis was performed to assess the affinity of IRI and Keap1–Nrf2 complex (Figure 4A). To validate the accuracy of molecular docking, we used quercetin (QUE), a known drug that activates Nrf2 by binding to the Keap1–Nrf2 complex [30,31], as a control. As shown in Figure 4B, quercetin exhibited a strong binding affinity to the Keap1–Nrf2 complex, with a minimum binding energy of −9.5 kcal/mol, which was consistent with previous findings [30]. Similarly, IRI displayed binding affinity with the Keap1–Nrf2 complex by forming hydrogen bonds with the 367th Glycine residue, 510th Alanine residue, and 465th and 512nd Valine residue, with a minimum binding energy of −9.5 kcal/mol (Figure 4B). These results indicate that IRI binds well with Keap1–Nrf2 in silico. The total protein expressions of Nrf2 and heme oxygenase-1 (HO-1) were significantly upregulated after IRI treatment (Figure 4C,D). Additionally, IF staining results showed that IRI treatment increased Nrf2 nuclear translocation, with the highest level observed under combined intervention with H_2_O_2_ and IRI (Figure 4E). Finally, a DCFH-DA kit was used to evaluate the effects of IRI on ROS generation. As shown in Figure 4F, the green fluorescence intensity was elevated after H_2_O_2_ stimulation and was significantly reduced following IRI treatment (Figure 4F).

### 2.5. Inhibition of Nrf2 Partially Diminished the Protective Effects of IRI

We utilized ML385, a specific Nrf2 inhibitor [32], to investigate whether IRI achieved its chondroprotective effects through activating Nrf2. The Western blot results indicated that ML385 effectively decreased the protein level of Nrf2 and partially blocked the therapeutic effects of IRI on H_2_O_2_-induced inflammation and cartilage degeneration (Figure 5A,B). IF staining for MMP13 further supported these findings (Figure 5C). Furthermore, inhibition of Nrf2 attenuated the anti-apoptotic effects of IRI against H_2_O_2_-induced chondrocyte apoptosis (Figure 5D,E).

### 2.6. IRI Effectively Delayed OA Progression In Vivo

To further investigate the chondroprotective effects of IRI on OA progression in vivo, a DMM-induced OA model was established in rats (Figure 6A). Micro-CT reconstruction of knee joints indicated a significant increase in osteophyte formation after DMM surgery compared to the sham group, whereas regular IRI injections reduced these morphological alterations (Figure 6B). The quantitative analysis of related parameters (BV/TV, Tb.N, Tb.Th, and Tb.Sp) revealed that IRI treatment decreased bone loss and suppressed subchondral bone remodeling (Figure 6C–F). In addition, the results of histological staining assay showed severe cartilage deformity, damage, and cartilage matrix loss in the DMM group. Conversely, IRI injections effectively alleviated the cartilage destruction (Figure 7A). The OARSI scores also supported this finding (Figure 7B). Immunohistochemistry (IHC) staining revealed that the rate of MMP13-positive cells increased after DMM surgery, while the rate of COL2-positive cells decreased. However, these results were reversed by IRI treatment (Figure 7C–F). In summary, these findings demonstrated that IRI exhibited chondroprotective effects and delayed OA progression in animals.

## 3. Discussion

IRI, which has anti-inflammatory and antioxidant activities, has shown therapeutic effect on various diseases. However, other studies regarding its potential for ameliorating OA progression are currently unavailable. Herein, we identified that IRI can inhibit inflammation and restore the balance between anabolism and catabolism in chondrocytes. Ample studies have reported that IRI can not only inhibit the production of inflammatory cytokines but also downregulate the protein expression of iNOS and COX2 [11,33]. In accordance with previous studies, our results also indicated that IRI significantly suppressed the expression of iNOS and COX-2 proteins in chondrocytes. Low-grade inflammation is involved in the pathogenesis of OA, and inflammatory mediators produced by cartilage also contribute to OA development [34,35]. Hence, anti-inflammatory IRI alleviated OA progression through decreasing the production of inflammatory mediators. In addition, IRI demonstrated potential in inhibiting matrix degradation. Zhang et al. identified that IRI could promote the expression of collagen II and aggrecan while inhibiting the upregulation of matrix metalloproteinases (MMPs) in TNF-α-stimulated nucleus pulposus cells [29]. Similarly, we found that IRI also promoted matrix synthesis while inhibiting matrix degradation in chondrocytes. This result indicates that IRI may be able to mitigate OA progression through inhibiting cartilage matrix degradation. 

Abnormal ROS accumulation promotes cell apoptosis and contributes to disease development [36]. Excess ROS levels are also involved in OA development, as oxidative stress resulting from overproduced ROS leads to inhibited chondrocyte proliferation and cellular apoptosis, eventually causing cartilage damage [37,38]. Nrf2, an important transcription factor in cellular redox balance, protects cartilage from degeneration through regulating downstream effectors like NAD (P)H:quinone oxidoreductase 1 [39], heme oxygenase-1 (HO-1) [40], and glutathione peroxidase 4 [41]. In addition to regulating redox homeostasis, Nrf2 also affects macrophage polarization by modulating autophagy and suppressing NF-κB nuclear translocation [42]. Wei et al. highlighted the application of Nrf2/HO-1-mediated metabolic reprogramming in the treatment of aplastic anemia [43]. Recent studies further emphasize the role of the Nrf2-HO-1 pathway in mitigating inflammation and cell damage caused by ferroptosis, suggesting its broad therapeutic potential [44,45]. Previous studies have demonstrated that IRI contributed to inflammation resolution through regulating the Nrf2/HO-1 pathway [11]. Hence, we believed that IRI alleviates OA progression through the Nrf2/HO-1 pathway. Experiments using a specific Nrf2 inhibitor (ML385) were carried out to further verify this conclusion. We observed that ML385 dampened the therapeutic effects of IRI on osteoarthritic chondrocytes. These results highlighted that IRI prevents cartilage from degeneration through regulating Nrf2/HO-1 pathway in chondrocytes.

Various flavonoid compounds have been reported to modulate redox homeostasis and exert therapeutic effects through the activation of Nrf2. For example, Gao et al. reported that a natural flavonoid, Icariside II, exerts neuroprotective effects in cases of an ischemic stroke by targeting and activating Nrf2 [46]. Another study found that Galangin reverses H_2_O_2_-induced fibroblast senescence through the Nrf2 signaling pathway [47]. Flavonoid-mediated Nrf2 activation also play an important role in the treatment of OA. Chen et al. found that Rhoifolin activates Nrf2 to improve chondrocyte senescence [48]. Baicalein enhances Nrf2 nuclear translocation to inhibit chondrocyte ferroptosis [49]. However, the role of IRI, a potent anti-inflammatory and antioxidant flavonoid, in osteoarthritis (OA) has not yet been studied. In this study, we demonstrated that IRI effectively inhibited inflammation, eliminated overproduced ROS in chondrocytes, and reduced chondrocyte apoptosis caused by H_2_O_2_-induced oxidative stress through activating the Nr2/HO-1 pathway.

Keap1 is known to bind Nrf2, leading to its ubiquitination and subsequent proteasomal degradation, thereby acting as a negative regulator of the Nrf2 signaling pathway [50]. Molecular docking analysis in this study sought to understand whether IRI could competitively bind to Keap1, thereby preventing Nrf2 degradation and activating the Nrf2 pathway. Molecular docking was employed to provide a detailed, atomistic understanding of IRI’s interaction with the Keap1–Nrf2 complex, complementing the in vivo and in vitro experiments. To validate our docking results, we employed two key strategies. First, quercetin, a known high-affinity ligand for the Keap–Nrf2 complex, was used as a positive control. Our analysis revealed that the minimum binding energy of IRI with the Keap1–Nrf2 complex is comparable to QUE. Next, experimental validation through Western blotting and immunofluorescence assays confirmed that IRI treatment increases Nrf2 expression and facilitates its nuclear translocation. Overall, the molecular docking results support the in vivo and in vitro findings, confirming that IRI exerts its anti-inflammatory and antioxidant therapeutic effects in OA through the activation of Nrf2.

## 4. Materials Methods

### 4.1. Chondrocyte Isolation and Culture

Chondrocytes were obtained from the knee joints of 6-day-old Sprague Dawley (SD) rats. After removing the surrounding tissues, cartilage was dissected and digested with 0.25% trypsin for 30 min. Subsequently, trypsin was removed, and the cartilage tissue was incubated with 0.25% type II collagenase solution at 37 °C for 6 h. The chondrocytes were then collected by centrifugation at 1500 rpm, resuspended in a DMEM/F12 medium (Hyclone, Logan, UT, USA), supplemented with 10% fetal bovine serum (FBS; Gibco, Grand Island, NY, USA) and 1% penicillin/streptomycin (Boster, Wuhan, China), and cultured in a 37 °C incubator with 5% CO_2_.

### 4.2. Cell Viability Analysis

Cell Counting Kit-8 (CCK-8; MedChemExpress, Monmouth Junction, NJ, USA) and Live/Dead staining assays were used to assess chondrocyte viability. For the CCK-8 assay, chondrocytes (approximately 10,000 cells per well) were seeded into a 96-well plate and stimulated with various concentrations of IRI (0, 5, 10, 20, 40, and 80 μM). After 24 h, 100 μL of CCK-8 reagent was added to each well and incubated for one hour. The optical density value of the cells was then measured at 450 nm using a microplate reader (BioTek, Winooski, VT, USA). For Live/Dead staining, chondrocytes were incubated with Calcein/PI Cell Viability/Cytotoxicity Reagent (Beyotime, Shanghai, China) for 30 min and visualized under a fluorescence microscope (Nikon, Tokyo, Japan).

### 4.3. Western Blotting

Following three washes with phosphate-buffered saline (PBS; Boster, Wuhan, China), chondrocytes were lysed with enhanced RIPA buffer (Boster) containing a 1 mM protease and phosphatase inhibitor cocktail (MedChemExpress) for 30 min. After determining the concentration of the protein samples using a BCA protein assay kit (Beyotime), approximately 20 μg of protein samples were separated by 10% or 12% SDS-PAGE gel electrophoresis and subsequently transferred onto the 0.2 μm polyvinylidene difluoride (PVDF; Millipore, Temecula, CA, USA) membranes. Then, 5% bovine serum albumin (BSA; BioFroxx, Heidelberg, Germany) was utilized to block the obtained membranes for 1 h. The blocked membranes were incubated with corresponding primary antibodies at 4 °C for 16 h. After rinsing with TBST solution for 10 min three times, the bands were incubated with corresponding secondary antibodies (Boster) for 1 h at room temperature. The target protein bands were visualized using an Image Lab System (BioRad, Hercules, CA, USA) after being immersed in ECL chemiluminescence reagent (Thermo Scientific, Agawam, MA, USA).

### 4.4. RNA Extraction and Real-Time Quantitative Polymerase Chain Reaction (RT-qPCR)

Total RNA was extracted and purified from chondrocytes using the E.Z.N.A.^®^ total RNA Kit I (Omega Bio-Tek, Norcross, GA, USA) according to the manufacturer’s protocol. After measuring the purity and concentration of the obtained RNA, 1 μg of RNA was reverse-transcribed into complementary DNA (cDNA) using the HiScript II cDNA Kit (Vazyme, Nanjing, China). The cDNA was then amplified and quantified using the RT-qPCR detection system (BioRad). Relative target mRNA levels were normalized to the level of *β-actin*. 

### 4.5. Immunofluorescence (IF) Staining

An appropriate number (20,000 cells/well) of chondrocytes were seeded onto a 24-well plate and subjected to various treatments for 24 h. After fixation with 4% paraformaldehyde for 20 min and membrane permeabilization with 0.1% Triton X-100 for 10 min, cells were blocked with 1% BSA for 1 h. Then, the corresponding primary antibodies (MMP13, COL2, iNOS, and Nrf2, 1:200) were incubated with the cells overnight at 4 °C. On the following day, the cells were incubated with the corresponding Cy3-conjugated (red) or FITC-conjugated (green) secondary antibodies (Boster) for 1 h at room temperature. The cell nuclei were stained with DAPI reagent for 5 min. Quantitative analysis of IF staining intensity was performed using ImageJ software (version 1.53v, National Institutes of Health, Bethesda, MD, USA).

### 4.6. Flow Cytometry Analysis of Cell Apoptosis

Chondrocytes were collected using trypsin after receiving different treatments for 24 h. The cells were resuspended in 500 μL of binding buffer. Then, Annexin V-APC and 7-AAD were added to stain the cells, according to the manufacturer’s protocol, for 10 min. Finally, apoptotic cells were detected and analyzed in a flow cytometer (Beckman Coulter, Brea, CA, USA).

### 4.7. TdT-Mediated dUTP Nick-End Labeling (TUNEL) Staining Assay

The TUNEL apoptosis detection kit (Biosharp, Hefei, China) was employed to assess chondrocyte apoptosis. After various treatments for 24 h, cells in a 96-well plate were fixed and permeabilized as mentioned above. Subsequently, 50 μL of reagent prepared according to the manufacturer’s instructions was added to each well and incubated for 60 min under light-shielded conditions. Following this, the chondrocytes were incubated with streptavidin-FITC reagent for 30 min to label the TdT enzyme. The cell nuclei were stained with DAPI reagent for 5 min. 

### 4.8. EdU Assay

The EdU assay (Beyotime) was employed to evaluate cell proliferation. Chondrocytes were seeded into a 96-well plate, and then 100 μL of EdU reagent (10 μM) was added to each well and incubated for 2 h. Following fixation and permeabilization, the cells were incubated in the dark with a staining solution for 1 h, followed by the addition of Hoechst 33342 for nuclear staining for 5 min. Finally, the stained cells were imaged under a fluorescence microscope.

### 4.9. ROS Detection

The ROS detection kit (MedChemExpress) was used to assess cellular ROS levels. In brief, chondrocytes were incubated with 100 μM of H_2_O_2_ and/or IRI for 24 h and then incubated with a DCFH-DA (10 μM) probe for 20 min, followed by washing three times with culture medium. Subsequently, the cells were incubated in the dark with Hoechst 33342 for 5 min and photographed under a fluorescence microscope.

### 4.10. Molecular Docking Analysis

The two-dimensional (2D) structure of IRI or QUE was obtained from the PubChem database (https://pubchem.ncbi.nlm.nih.gov/) and converted into a three-dimensional (3D) structure using Chem3D 19.0 software. These 3D structures were then optimized to their lowest energy conformations. The protein structure of a Keap1–Nrf2 complex was downloaded from the PDB database (PDBID: 4XMB; http://www.wwpdb.org/). The preparation of the Keap1–Nrf2 complex involved the removal of water molecules and the addition of hydrogen atoms, following instructions for the docking grid box using AutoDockTools 1.5.7. Molecular docking was performed with AutoDock Vina 1.2.3 software, generating 20 docking modes per run to identify the optimal binding conformations. The binding energy of these 20 docking modes were then calculated. Finally, the optimal binding mode was visualized using PyMOL 3.2, which provided detailed information on the binding sites, the number of hydrogen bonds, and the lengths of these hydrogen bonds. Generally, a binding energy below −7.0 kcal/mol suggests a strong binding affinity.

### 4.11. In Vivo Experiments

All animal experiments were approved and instructed by the Ethic Committee of the Experimental Animal Center of Huazhong University of Science and Technology (No. 3998, Wuhan, China). Eighteen male SD rats (6 weeks old) were purchased from Shulaibao Biotechnology Company (Wuhan, China). Following a two-week acclimatization period, these rats were randomly divided into three groups (*n* = 6): Sham group, DMM group, and DMM + IRI group. Rats in the sham group underwent only an incision of the medial joint capsule. In the DMM and DMM + IRI group, rats underwent destabilization of the medial meniscus (DMM) surgery on the right joint under 3% pentobarbital sodium general anesthesia (40 mg/kg). Two weeks post-surgery, the rats in the DMM + IRI group received a 10 µL intra-articular injection of IRI (1 mg/kg), while the rats in the sham and DMM groups were administrated with 10 μL of PBS. The injection procedures were conducted twice weekly for a duration of 4 weeks. After that, all rats were euthanized, and their right joints were collected for further experiments.

### 4.12. Micro-CT Evaluation

The harvested knee joints were fixed with 4% formaldehyde for 72 h. The fixed samples were then scanned and analyzed using a micro-CT 50 scanner (Scanco Medical AG, Wangen-Brüttisellen, Switzerland). Scanco Medical software v6.1 was employed to reconstruct and obtain 3D images of knee joints. Bone morphological parameters, including bone volume/tissue volume fraction (BV/TV), trabecular thickness (Tb.Th), trabecular number (Tb.N), and trabecular separation (Tb.Sp) were analyzed to assess bone mass and subchondral bone structure.

### 4.13. Histological Staining and Immunohistochemistry

The obtained knee joints were fixed and then decalcified with decalcifying solution (Boster) for 2 months. Subsequently, the samples were embedded and cut into 5 μm paraffin sections. Hematoxylin–Eosin (H&E; Servicebio, Wuhan, China) staining, toluidine blue assay (TB; Servicebio), and safranin O-fast green (SO-FG; Beyotime) staining were performed for histological analysis. The Osteoarthritis Research Society International (OARSI) score was applied for the evaluation of the cartilage morphology. For immunohistochemistry (IHC) staining, the slices were incubated with the primary antibodies (anti-MMP13 and anti-Aggrecan), corresponding secondary antibody, and DAB in sequence. The bright-field images were visualized and photographed under a microscope.

### 4.14. Statistical Analysis

GraphPad Prism 9 software was employed to perform statistical analysis. All experiments in this study were conducted at least three times and the data were shown as mean ± standard deviation (SD). The differences of experimental data between different groups were compared using one-way analysis of variance (ANOVA). In addition, a Kruskal-Wallis H test was used to evaluate the OARSI score. A *p* < 0.05 was considered statistically significant.

## 5. Conclusions

In this study, we found that IRI could alleviate OA development in vitro and in vivo. IRI significantly suppressed the production of inflammatory mediators and the expression of MMP13, thereby preventing cartilage from degradation. Additionally, IRI promoted synthesis of glycosaminoglycan and collagen, both of which are major components in the cartilage matrix. Finally, IRI reduced oxidative-stress-induced chondrocyte apoptosis, and its in vivo therapeutic effect was verified with a rat OA model. In conclusion, we identified IRI as a potential option for OA treatment and revealed the mechanism underlying the therapeutic effects of IRI on OA. 

## Figures and Tables

**Figure 1 pharmaceuticals-17-01268-f001:**
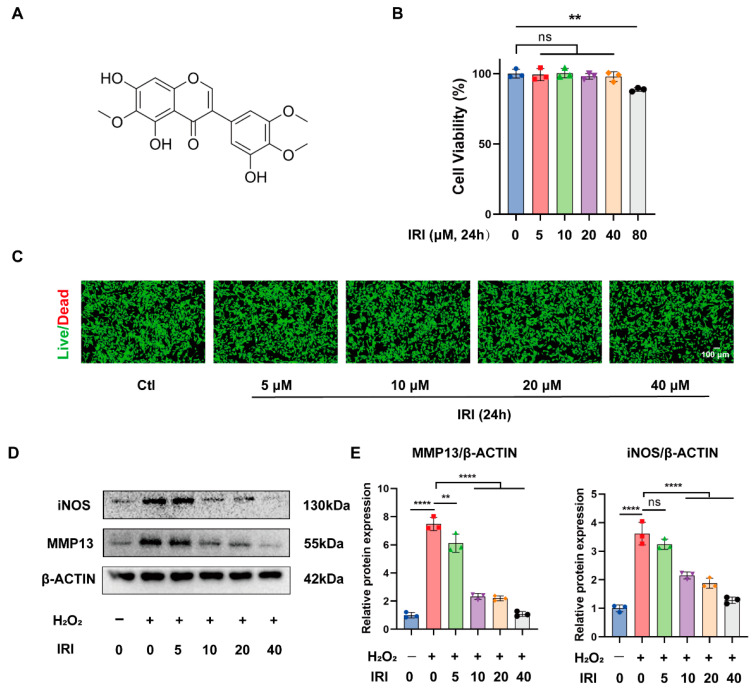
Effects of IRI on chondrocyte viability and selection of the optimal treatment concentration of IRI. (**A**) The chemical structure of IRI. (**B**) Chondrocyte viability evaluation of IRI at 0, 5, 10, 20, 40, and 80 μM for 24 h. Data are presented as mean ± SD; *n* = 3/group. Statistical significance markers are as follows: ** *p* < 0.01; ns, not significance, calculated by one-way ANOVA, compared with the Ctl group. (**C**) Live/Dead staining of chondrocytes after stimulation with IRI at different concentrations (0, 5, 10, 20, and 40 μM) for 24 h (green, live cells; red, nucleus of dead cells). (**D**) The protein levels of MMP13 and iNOS in chondrocytes treated with 400 μM of H_2_O_2_ and IRI at different concentrations (0–40 μM) for 24 h. (**E**) Quantitative analysis of MMP13 and iNOS. Data are presented as mean ± SD; *n* = 3/group. Statistical significance markers are as follows: ** *p* < 0.01; **** *p* < 0.0001; ns, not significant, calculated by one-way ANOVA.

**Figure 2 pharmaceuticals-17-01268-f002:**
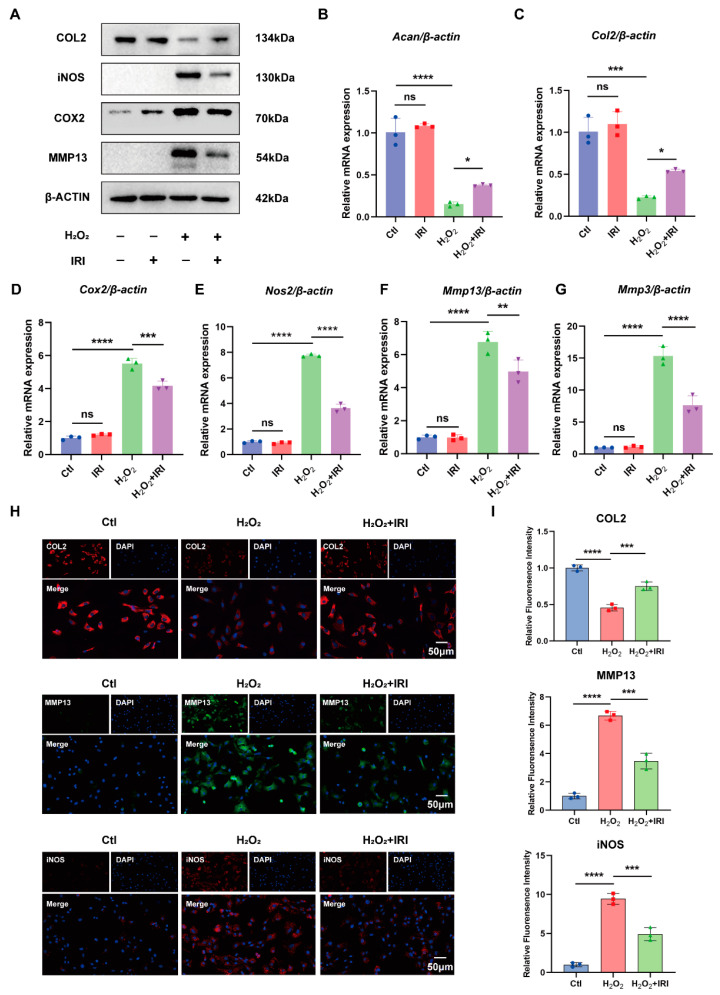
IRI inhibited the inflammation response and restored matrix homeostasis in the H_2_O_2_-stimulated chondrocytes. (**A**) The protein levels of COL2, iNOS, COX2, and MMP13 in chondrocytes treated with IRI and/or H_2_O_2_ for 24 h. (**B**–**G**) The relative mRNA levels of *Acan*, *Col2*, *Cox2*, *Nos2*, *Mmp13*, and *Mmp3* in chondrocytes treated with IRI and/or H_2_O_2_ for 24 h. Data are presented as mean ± SD; *n* = 3/group. Statistical significance markers are as follows: * *p* < 0.05; ** *p* < 0.01; *** *p* < 0.001; **** *p* < 0.0001; ns, not significant, calculated by one-way ANOVA. (**H**) The representative IF staining of COL2, MMP13, and iNOS in chondrocytes. (**I**) Quantitative analysis of fluorescence intensity. Data are presented as mean ± SD; *n* = 3/group. Significance levels are described as in (**B**–**G**).

**Figure 3 pharmaceuticals-17-01268-f003:**
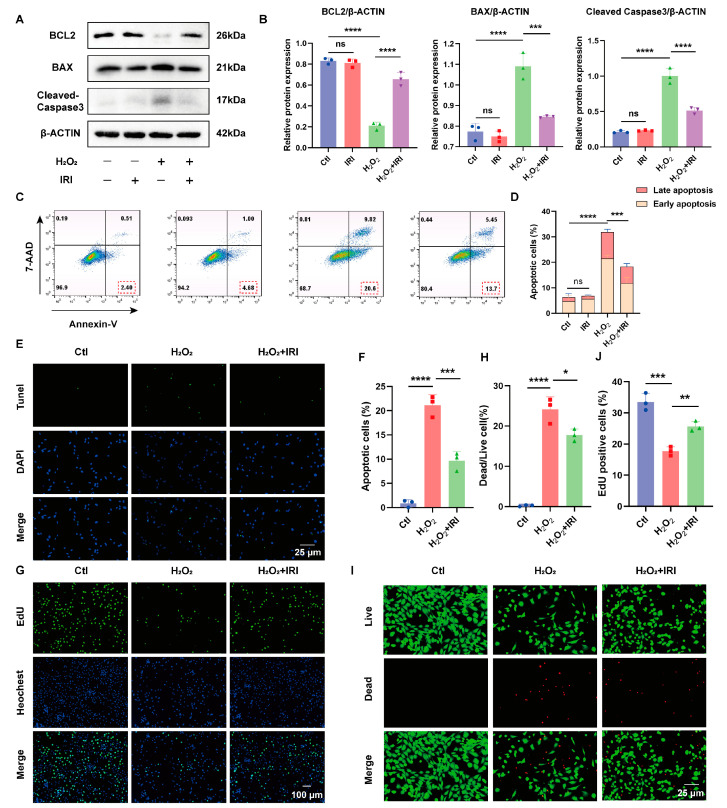
IRI attenuated apoptosis and promoted proliferation in the H_2_O_2_-induced chondrocytes. (**A**,**B**) The protein levels of BCL2, BAX, and cleaved caspase-3 in chondrocytes treated with IRI and/or H_2_O_2_ for 24 h. Data are presented as mean ± SD; *n* = 3/group. Statistical significance markers are as follows: * *p* < 0.05; ** *p* < 0.01; *** *p* < 0.001; **** *p* < 0.0001; ns, not significant, calculated by one-way ANOVA. (**C**,**D**) Flow cytometry analysis of chondrocyte apoptosis received various treatments. Data are presented as mean ± SD; *n* = 3/group. Significance levels are specified as in (**B**). (**E**) Representative images of TUNEL staining. (**F**) Percentage of apoptotic chondrocytes detected by TUNEL staining. Data are presented as mean ± SD; *n* = 3/group. Significance levels are specified as in (**B**). (**G**) Representative images of EdU staining. (**H**) Percentage of proliferative chondrocytes measured by EdU staining. Data are presented as mean ± SD; *n* = 3/group. Significance levels are specified as in (**B**). (**I**) Representative images of Live/Dead staining. (**J**) Percentage of dead chondrocytes detected by Live/Dead staining. Data are presented as mean ± SD; *n* = 3/group. Significance levels are specified as in (**B**).

**Figure 4 pharmaceuticals-17-01268-f004:**
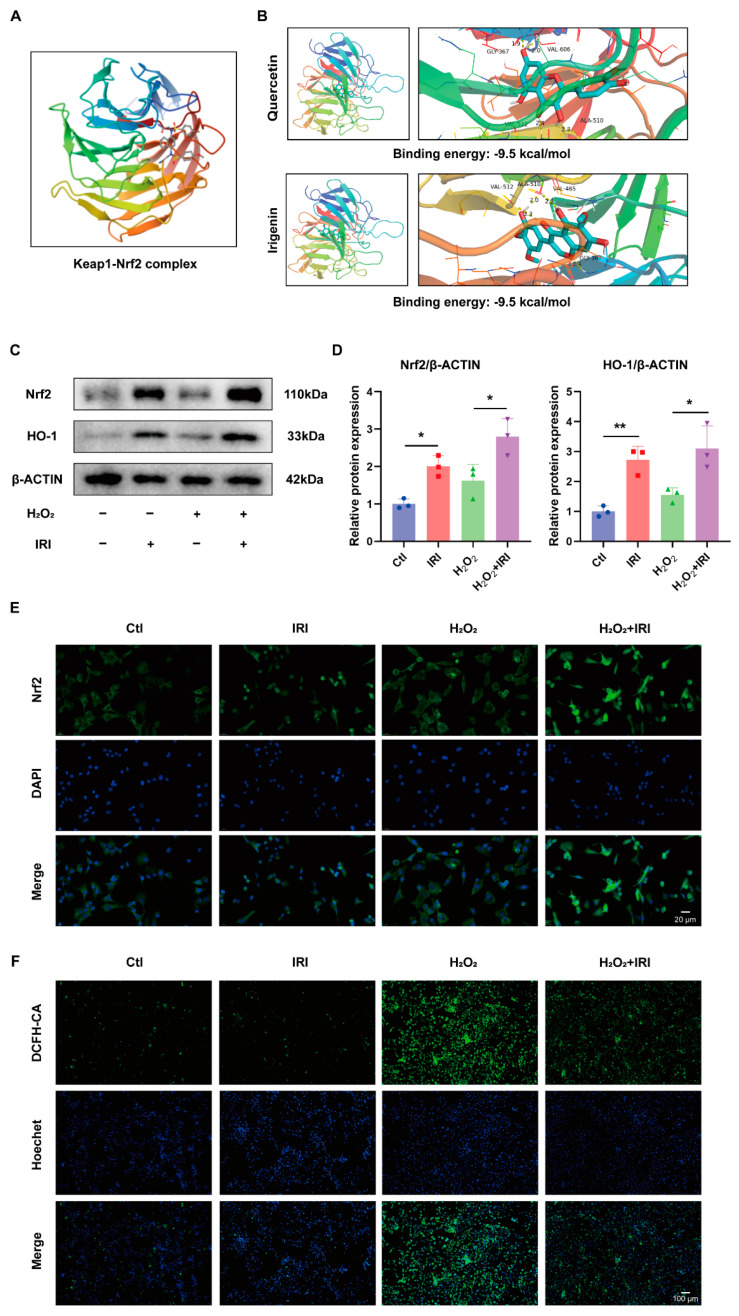
IRI decreased oxidative stress through activating the Nrf2/HO-1 axis in chondrocytes. (**A**) The crystal structure of the Keap1–Nrf2 complex. (**B**) The crystal structure of the docking analysis of IRI and QUE with the Keap1–Nrf2 complex. (**C**,**D**) The protein levels of Nrf2 and HO-1 in chondrocytes treated with IRI and/or H_2_O_2_ for 24 h. Data are presented as mean ± SD; *n* = 3/group. Statistical significance markers are as follows: * *p* < 0.05; ** *p* < 0.01, calculated by one-way ANOVA. (**E**) The representative IF images of Nrf2 in chondrocytes treated with IRI and/or H_2_O_2_ for 24 h. (**F**) The representative images of intracellular ROS detected by DCFH-DA fluorescent staining.

**Figure 5 pharmaceuticals-17-01268-f005:**
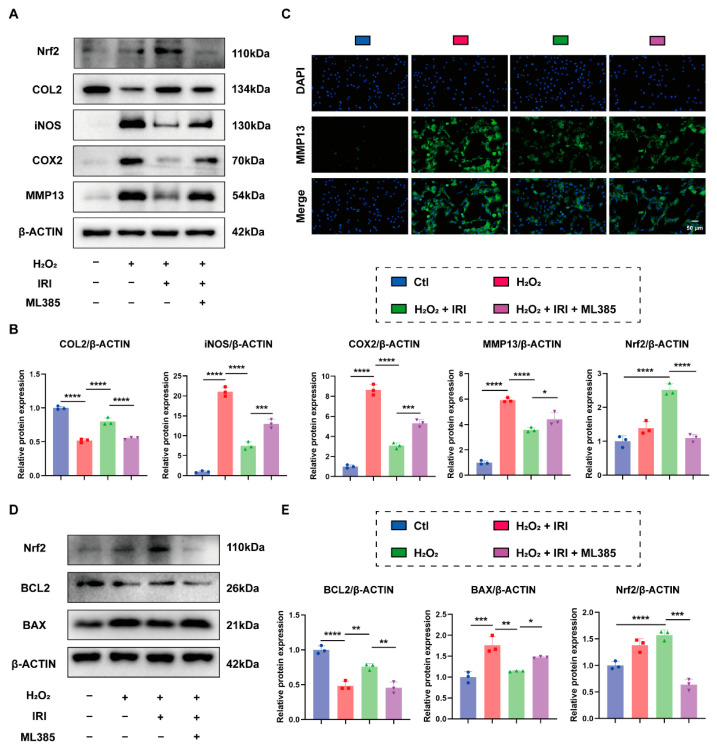
Inhibition of Nrf2 partially diminished the protective effects of IRI. (**A**,**B**) The protein levels of Nrf2, COL2, iNOS, COX2, and MMP13 in chondrocytes stimulated with various treatments. Data are presented as mean ± SD; *n* = 3/group. Statistical significance markers are as follows: * *p* < 0.05; ** *p* < 0.01; *** *p* < 0.001; **** *p* < 0.0001; ns, not significant, calculated by one-way ANOVA. (**C**) Representative IF images of MMP13 in chondrocytes. (**D**,**E**) The protein levels of Nrf2, BCL2, and BAX in chondrocytes stimulated with various treatments. Data are presented as mean ± SD; *n* = 3/group. Significance levels are specified as in (**B**).

**Figure 6 pharmaceuticals-17-01268-f006:**
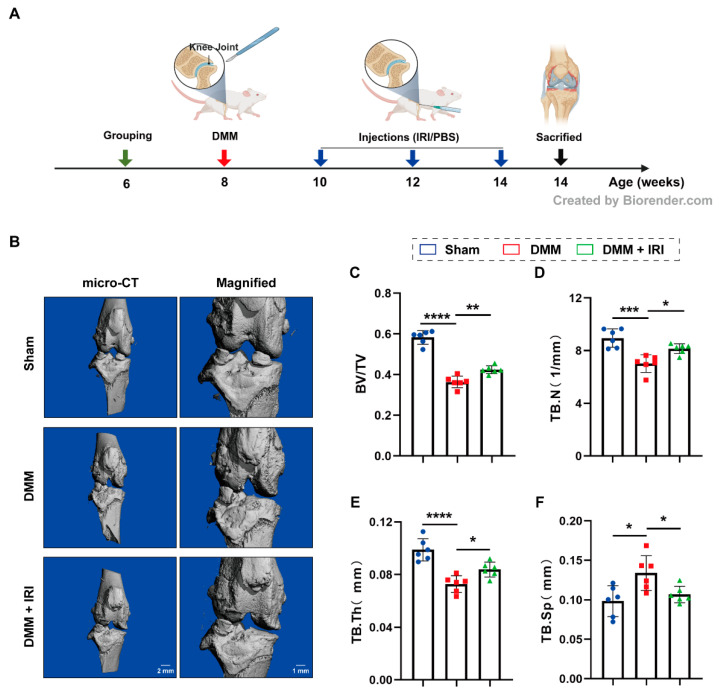
IRI effectively delayed OA progression in vivo. (**A**) Schematic diagram of intra-articular injection of IRI in a DMM-induced OA model. (**B**) The 3D reconstruction images of knee joints from rats in different groups. (**C**–**F**) Quantitative analysis of the related bone-morphological-parameters analysis (BV/TV, Tb.N, Tb.Th, and Tb.Sp). Data are presented as mean ± SD; *n* = 6/group. Statistical significance markers are as follows: * *p* < 0.05; ** *p* < 0.01; *** *p* < 0.001; **** *p* < 0.0001, calculated by one-way ANOVA.

**Figure 7 pharmaceuticals-17-01268-f007:**
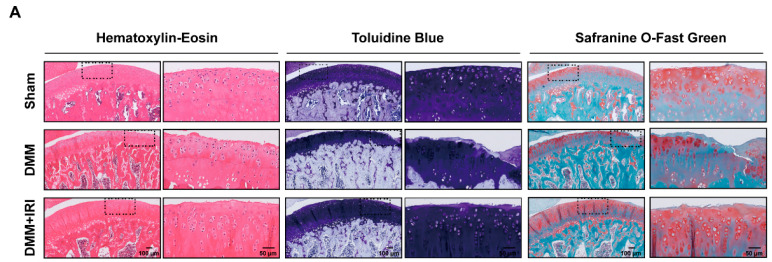
IRI effectively delayed OA progression in vivo. (**A**) H&E, TB and SO-FG staining images of articular cartilage. (**B**) Statistical analysis of OARSI scores. Data are presented as mean ± SD; *n* = 6/group. Statistical significance markers are as follows: *** *p* < 0.001; **** *p* < 0.0001, determined by the Kruskal-Wallis H test. (**C**,**D**) The representative images and quantitative analysis of Aggrecan level. Data are presented as mean ± SD; *n* = 6/group; Statistical significance markers are as follows: ** *p* < 0.01; *** *p* < 0.001; **** *p* < 0.0001, calculated by one-way ANOVA. (**E**,**F)** The representative images and quantitative analysis of MMP13 level. Data are presented as mean ± SD; *n* = 6/group. Significance levels are specified as in (**D**).

## Data Availability

The data that support the findings of this study are available from the corresponding author upon reasonable request.

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
