# Peer review of "Anti-Inflammatory and Antioxidant Effects of Irigenen Alleviate Osteoarthritis Progression through Nrf2/HO-1 Pathway"

_pharmaceuticals, 2024, doi:10.3390/ph17101268_

Round 1

Reviewer 1 Report

Comments and Suggestions for Authors

Dear Editor,

Thank you for the opportunity to review this manuscript. The study presents a well-structured investigation into the potential therapeutic effects of irigenin on osteoarthritis (OA) through the modulation of the Nrf2/HO-1 pathway. The research is thorough and contributes valuable insights to the field. Below, I provide some suggestions for minor revisions that I believe will enhance the clarity and impact of the paper.

  Methodological Details:

It is recommended to provide more detailed descriptions of the parameters and procedures used in the molecular docking analysis in the Methods section to ensure the reproducibility of the results by other researchers.

  Data Presentation:

While the data figures in the Results section are clear, it is suggested to include more detailed explanations of statistical significance markers in the figure legends to enhance the clarity and interpretability of the data.

  Discussion Enhancement:

The Discussion section could be enriched by exploring other potential regulatory mechanisms of the Nrf2/HO-1 pathway, providing a more comprehensive scientific discussion.

  Reference Updates:

It is advisable to update the reference list to ensure that the most recent and relevant studies, particularly those related to the Nrf2/HO-1 pathway and irigenin, are included.

  Language and Expression:

Although the scientific content is the primary focus, clear and fluent language is equally important. It is recommended that the authors review the manuscript for any grammatical errors or unclear expressions to improve the overall readability of the paper.

The study is well-designed, the results are clearly presented, and the discussion is thorough. Minor revisions are suggested before the paper is accepted for publication.

Author Response

Thank you for the opportunity to review this manuscript. The study presents a well-structured investigation into the potential therapeutic effects of irigenin on osteoarthritis (OA) through the modulation of the Nrf2/HO-1 pathway. The research is thorough and contributes valuable insights to the field. Below, I provide some suggestions for minor revisions that I believe will enhance the clarity and impact of the paper.

  1. Methodological Details:

It is recommended to provide more detailed descriptions of the parameters and procedures used in the molecular docking analysis in the Methods section to ensure the reproducibility of the results by other researchers.

Answer: Thank you for your valuable suggestions. Following your advice, the related descriptions of procedures and parameters in the molecular docking analysis were complemented in the revised manuscript (line 362-376).

  1. Data Presentation:

While the data figures in the Results section are clear, it is suggested to include more detailed explanations of statistical significance markers in the figure legends to enhance the clarity and interpretability of the data.

Answer: Thank you for your suggestions. Based on your feedback, we have revised the figure legends to provide more detailed explanations of the statistical significance markers (line 621-704).

  1. Discussion Enhancement:

The Discussion section could be enriched by exploring other potential regulatory mechanisms of the Nrf2/HO-1 pathway, providing a more comprehensive scientific discussion.

Answer: Thank you for your valuable suggestions. The related discussion of the other regulatory mechanisms of Nrf2/HO-1 pathway was added to the revised manuscript (line 228-240).

  1. Reference Updates:

It is advisable to update the reference list to ensure that the most recent and relevant studies, particularly those related to the Nrf2/HO-1 pathway and irigenin, are included.

Answer: Thank you for your comments. We have updated the references based on your feedback to include the latest and relevant research on Nrf2/HO-1 pathway and Irigenin, including references such as 11, 33, 42-49.

  1. Language and Expression:

Although the scientific content is the primary focus, clear and fluent language is equally important. It is recommended that the authors review the manuscript for any grammatical errors or unclear expressions to improve the overall readability of the paper.

Answer: Thank you for your suggestions. We have carefully reviewed the manuscript to correct the grammatical errors and unclear expressions in the manuscript. The corresponding modifications were highlighted in red.

The study is well-designed, the results are clearly presented, and the discussion is thorough. Minor revisions are suggested before the paper is accepted for publication.

Reviewer 2 Report

Comments and Suggestions for Authors

The study presents compelling evidence for the efficacy of Irigenin (IRI) in alleviating osteoarthritis (OA) progression through its anti-inflammatory and antioxidant properties. The research thoroughly demonstrates that IRI can restore chondrocyte matrix homeostasis and suppress oxidative stress-induced inflammation and apoptosis, primarily through the activation of the Nrf2/HO-1 pathway. The in vivo and in vitro experiments provide robust support for IRI's potential as a therapeutic agent, particularly highlighting its ability to reduce cartilage degeneration and promote chondrocyte survival. However, while the results are promising, the study would benefit from a more detailed exploration of the long-term effects of IRI, as well as its efficacy compared to existing treatments for OA. Moreover, the reliance on a single pathway for therapeutic efficacy raises concerns about potential resistance or limited effectiveness in more complex biological systems.

In conclusion, the study offers valuable insights into the therapeutic potential of IRI for OA treatment, yet it also opens avenues for further research to fully establish its clinical applicability. Future studies should aim to address the long-term outcomes of IRI treatment, its comparative effectiveness, and potential side effects in broader patient populations. Additionally, investigating the interplay between the Nrf2/HO-1 pathway and other signaling pathways involved in OA could provide a more comprehensive understanding of IRI's mechanism of action.

Questions:

1. How does the therapeutic efficacy of Irigenin compare to current standard treatments for osteoarthritis?

2. What are the potential long-term side effects of Irigenin treatment in vivo?

3. Could the activation of the Nrf2/HO-1 pathway by Irigenin lead to resistance or reduced effectiveness over time?

4. How might Irigenin interact with other signaling pathways involved in osteoarthritis progression?

Comments on the Quality of English Language

The English language is clear with minor grammatical and stylistic improvements needed

Author Response

The study presents compelling evidence for the efficacy of Irigenin (IRI) in alleviating osteoarthritis (OA) progression through its anti-inflammatory and antioxidant properties. The research thoroughly demonstrates that IRI can restore chondrocyte matrix homeostasis and suppress oxidative stress-induced inflammation and apoptosis, primarily through the activation of the Nrf2/HO-1 pathway. The in vivo and in vitro experiments provide robust support for IRI's potential as a therapeutic agent, particularly highlighting its ability to reduce cartilage degeneration and promote chondrocyte survival. However, while the results are promising, the study would benefit from a more detailed exploration of the long-term effects of IRI, as well as its efficacy compared to existing treatments for OA. Moreover, the reliance on a single pathway for therapeutic efficacy raises concerns about potential resistance or limited effectiveness in more complex biological systems.

In conclusion, the study offers valuable insights into the therapeutic potential of IRI for OA treatment, yet it also opens avenues for further research to fully establish its clinical applicability. Future studies should aim to address the long-term outcomes of IRI treatment, its comparative effectiveness, and potential side effects in broader patient populations. Additionally, investigating the interplay between the Nrf2/HO-1 pathway and other signaling pathways involved in OA could provide a more comprehensive understanding of IRI's mechanism of action.

Questions:

  1. How does the therapeutic efficacy of Irigenin compare to current standard treatments for osteoarthritis?

Answer: Thank you for your insightful question. The ultimate goal of basic research is indeed to provide new strategies for clinical treatment. The present study focuses on exploring the potential therapeutic effects of IRI in OA using cellular and animal models. At this stage, we lack clinical trial data and therefore cannot directly compare the efficacy of IRI with existing standard treatments for OA. However, as you mentioned, our research suggests that IRI could be a promising adjunctive treatment option for OA, which merits further clinical investigation. We recognize the importance of clinical trials in validating these results and are committed to further research to establish a clearer comparison with current standard therapies. We appreciate your understanding and look forward to advancing this research in future studies.

  1. What are the potential long-term side effects of Irigenin treatment in vivo?

Answer: Thank you for your valuable comments. Based on our current animal studies, where IRI was administered via intra-articular injection into the knee joint for a duration of four weeks, we have not observed any significant adverse effects. The animals tolerated the treatment well, with no noticeable toxicity or major side effects recorded during this period. Although our study did not investigate potential long-term side effects, existing literatures provide some insights into the short-term safety of IRI similar to our research. For instance, a study administering IRI (10 or 20 mg/kg/day) daily for 5 weeks reported no adverse effects [1]. Another study found that daily oral administration of 20 mg/kg IRI did not affect mice’s weight or show side effects [2]. These findings suggest that short-term use of IRI is relatively safe. Nevertheless, potential long-term side effects of Irigenin remain unexplored in our study and the existing literature. Further research is needed to comprehensively evaluate the long-term safety profile of IRI. We are committed to investigating these aspects in future studies to better understand the overall safety and efficacy of IRI.

[1]. Guo L, Zheng X, Wang E, Jia X, Wang G, Wen J. Irigenin treatment alleviates doxorubicin (DOX)-induced cardiotoxicity by suppressing apoptosis, inflammation and oxidative stress via the increase of miR-425. Biomed Pharmacother. 2020;125:109784. doi:10.1016/j.biopha.2019.109784

[2]. Xu J, Sun S, Zhang W, et al. Irigenin inhibits glioblastoma progression through suppressing YAP/β-catenin signaling. Front Pharmacol. 2022;13:1027577. Published 2022 Nov 30. doi:10.3389/fphar.2022.1027577

  1. Could the activation of the Nrf2/HO-1 pathway by Irigenin lead to resistance or reduced effectiveness over time?

Answer: Thank you for your insightful comments. We acknowledge that drug resistance arising from prolonged treatment is indeed an important issue that warrants attention. Based on our current data and literature review [1-3], we have not observed indications of drug resistance or decreased efficacy of Irigenin or Nrf2/HO-1 pathway activation in the short to medium-term studies. Nevertheless, the possibility of developing resistance or reduced efficacy with long-term use of Irigenin is a valid concern. As you suggested, our current research paves the way for further studies. We will include additional investigations in our future study to explore the long-term efficacy and potential resistance associated with Irigenin treatment. This will help establish a foundation for its clinical application.

[1]. Sun YY, Zhu HJ, Zhao RY, et al. Remote ischemic conditioning attenuates oxidative stress and inflammation via the Nrf2/HO-1 pathway in MCAO mice. Redox Biol. 2023;66:102852. doi:10.1016/j.redox.2023.102852

[2]. Wang H, Yu X, Liu D, et al. VDR Activation Attenuates Renal Tubular Epithelial Cell Ferroptosis by Regulating Nrf2/HO-1 Signaling Pathway in Diabetic Nephropathy. Adv Sci (Weinh). 2024;11(10):e2305563. doi:10.1002/advs.202305563

[3]. Guo L, Zheng X, Wang E, Jia X, Wang G, Wen J. Irigenin treatment alleviates doxorubicin (DOX)-induced cardiotoxicity by suppressing apoptosis, inflammation and oxidative stress via the increase of miR-425. Biomed Pharmacother. 2020;125:109784. doi:10.1016/j.biopha.2019.109784

  1. How might Irigenin interact with other signaling pathways involved in osteoarthritis progression?

Answer: Thank you for your valuable question. In the previous studies, Irigenin has demonstrated therapeutic effects in various diseases, with one of the well-reported mechanisms being its ability to activate the Nrf2/HO-1 pathway, which enhances cellular resistance to oxidative stress and inflammation [1-2]. Given that OA is a degenerative metabolic disease closely related to aging, oxidative stress and low-grade inflammation play significant roles in its pathogenesis [3]. The preclinical and clinical studies have shown that oxidative stress contributes to the development of OA by inducing an aging-like phenotype in chondrocytes [4-5]. Therefore, in our study of Irigenin’s mechanism in treating OA, our primary focus is on the activation of the Nrf2/HO-1 pathway. As you suggested, exploring how Irigenin might interact with other signaling pathways is crucial. For example, the MAPK pathway, an important pathway that promotes OA progression, is one of the mechanisms through which Irigenin exerts its pharmacological effects [6]. In future research, we will further investigate how Irigenin interacts with other signaling pathways to provide a more comprehensive understanding of its therapeutic potential in OA.

[1]. Guo F, Wang X, Liu X. Protective effects of irigenin against 1-methyl-4-phenylpyridinium-induced neurotoxicity through regulating the Keap1/Nrf2 pathway. Phytother Res. 2021;35(3):1585-1596. doi:10.1002/ptr.6926

[2]. Zhang Q, Xue T, Guan J, et al. Irigenin alleviates angiotensin II-induced oxidative stress and apoptosis in HUVEC cells by activating Nrf2 pathway. Drug Dev Res. 2021;82(7):999-1007. doi:10.1002/ddr.21802

[3]. Liu, Liang et al. “The physiological metabolite α-ketoglutarate ameliorates osteoarthritis by regulating mitophagy and oxidative stress.” Redox biology vol. 62 (2023): 102663. doi:10.1016/j.redox.2023.102663

[4]. Hui W, Young DA, Rowan AD, Xu X, Cawston TE, Proctor CJ. Oxidative changes and signalling pathways are pivotal in initiating age-related changes in articular cartilage. Ann Rheum Dis. 2016;75(2):449-458. doi:10.1136/annrheumdis-2014-206295

[5]. Yudoh K, Nguyen vT, Nakamura H, Hongo-Masuko K, Kato T, Nishioka K. Potential involvement of oxidative stress in cartilage senescence and development of osteoarthritis: oxidative stress induces chondrocyte telomere instability and downregulation of chondrocyte function. Arthritis Res Ther. 2005;7(2):R380-R391. doi:10.1186/ar1499

[6]. Liu D, Wang Q, Yuan W, Wang Q. Irigenin attenuates lipopolysaccharide-induced acute lung injury by inactivating the mitogen-activated protein kinase (MAPK) signaling pathway. Hum Exp Toxicol. 2023;42:9603271231155098. doi:10.1177/09603271231155098

Comments on the Quality of English Language

The English language is clear with minor grammatical and stylistic improvements needed

Answer: Thank you for your feedback. We have carefully reviewed the manuscript and made the necessary grammatical and stylistic corrections to improve clarity.

Reviewer 3 Report

Comments and Suggestions for Authors

The authors conducted a comprehensive study involving in silico, in vitro, and in vivo approaches to investigate the role of irigenin in an osteoarthritis model. The study delves into mechanistic aspects and contributes to advances in the field. However, while many flavonoids with similar structures are known to affect Nrf2, the authors did not discuss this aspect. The manuscript is suitable for publication in Pharmaceuticals after addressing the following points for further exploration or revision:

(1) Some expressions are redundant, such as “natural plants” and “molecular structure.”

(2) The Y-axis of Figure 1b should be expressed as a percentage of survival.

(3) The font size of the axes in the graphs is very small and needs to be increased.

(4) The Discussion and Conclusion sections are currently combined; it is strongly recommended to separate these into distinct sections.

(5) The authors should provide a clearer explanation for the absence of a control treatment for osteoarthritis. Additionally, the in silico investigations did not include a ligand for Nrf2. The molecular docking analysis is underexplored and lacks validation; a dedicated section on molecular docking is suggested.

(6) The role of others flavonoids on the investigated molecular target is poorly discussed. 

Author Response

The authors conducted a comprehensive study involving in silico, in vitro, and in vivo approaches to investigate the role of irigenin in an osteoarthritis model. The study delves into mechanistic aspects and contributes to advances in the field. However, while many flavonoids with similar structures are known to affect Nrf2, the authors did not discuss this aspect. The manuscript is suitable for publication in Pharmaceuticals after addressing the following points for further exploration or revision:

  1. Some expressions are redundant, such as “natural plants” and “molecular structure.”

Answer: Thank you for your feedback. We have carefully reviewed the manuscript and removed the redundant expressions.

  1. The Y-axis of Figure 1b should be expressed as a percentage of survival.

Answer: Thank you for your valuable suggestions. Following your advice, the Y-axis of Figure 1B has been modified to represent the percentage of survival (Figure 1B).

  1. The font size of the axes in the graphs is very small and needs to be increased.

Answer: Thank you for your feedback. We have increased the font size of the axes in the graphs to improve clarity. We hope these adjustments make the figures more readable.

  1. The Discussion and Conclusion sections are currently combined; it is strongly recommended to separate these into distinct sections.

Answer: Thank you for your valuable suggestions. Based on your advice and the requirements of the journal, we have separated the Discussion and Conclusion sections and adjusted the structure order of the manuscript (line 203-270; line420-428).

  1. The authors should provide a clearer explanation for the absence of a control treatment for osteoarthritis. Additionally, the in silico investigations did not include a ligand for Nrf2. The molecular docking analysis is underexplored and lacks validation; a dedicated section on molecular docking is suggested.

5.1 The authors should provide a clearer explanation for the absence of a control treatment for osteoarthritis.

Answer: Thank you for your insightful comments. We understand your concern regarding the absence of a control treatment for OA in our study. we aimed to investigate the therapeutic effects of Irigenin on OA both in vitro and in vivo as a potential treatment candidate and explore its underlying mechanism. As such, we did not include a positive control treatment group to directly observe the impact of Irigenin. However, we acknowledge that comparing its effects with current standard treatments, as well as assessing potential side effects, requires further investigation in future studies. we plan to incorporate a control treatment group in our future studies. This will allow us to compare the effects of Irigenin with existing treatments and provide a more comprehensive assessment of its clinical potential for OA treatment.

5.2 The in silico investigations did not include a ligand for Nrf2. The molecular docking analysis is underexplored and lacks validation.

Answer: Thank you for your suggestions. Based on your advice and a thorough literature review [1-4], we have expanded our in silico studies to include Keap1 as a ligand for Nrf2. We performed molecular docking analysis between Irigenin and the Keap1-Nrf2 complex (Figure 4A). Our results demonstrate that Irigenin exhibits a strong affinity for the Keap1-Nrf2 complex, indicating that IRI competitively binds with Keap1, thereby preventing Nrf2 degradation and activating the Nrf2 pathway (Figure 4B, line 159-167). To validate these molecular docking results, we used quercetin (QUE) as a positive control, given its known high affinity for the Keap1/Nrf2 complex [2, 5]. The analysis shows that the affinity of IRI for the Keap1-Nrf2 complex is comparable to that of QUE (Figure 4B). Additionally, the Western blot and immunofluorescence experiments also confirm that IRI treatment increases Nrf2 expression and promotes its nuclear translocation (Figure 4C-E). We hope these additions address your concerns and enhance the robustness of our study.

[1]. Li, Huiyi et al. “Kaempferol prevents acetaminophen-induced liver injury by suppressing hepatocyte ferroptosis via Nrf2 pathway activation.” Food & function vol. 14,4 1884-1896. 21 Feb. 2023, doi:10.1039/d2fo02716j

[2]. Luo, Xing et al. “A novel anti-atherosclerotic mechanism of quercetin: Competitive binding to KEAP1 via Arg483 to inhibit macrophage pyroptosis.” Redox biology vol. 57 (2022): 102511. doi:10.1016/j.redox.2022.102511

[3]. Deng, Xuehui et al. “Mangiferin attenuates osteoporosis by inhibiting osteoblastic ferroptosis through Keap1/Nrf2/SLC7A11/GPX4 pathway.” Phytomedicine : international journal of phytotherapy and phytopharmacology vol. 124 (2024): 155282. doi:10.1016/j.phymed.2023.155282

[4]. Song, Jiayu et al. “Vitexin attenuates chronic kidney disease by inhibiting renal tubular epithelial cell ferroptosis via NRF2 activation.” Molecular medicine (Cambridge, Mass.) vol. 29,1 147. 27 Oct. 2023, doi:10.1186/s10020-023-00735-1

[5]. Shao, Z et al. “Senolytic agent Quercetin ameliorates intervertebral disc degeneration via the Nrf2/NF-κB axis.” Osteoarthritis and cartilage vol. 29,3 (2021): 413-422. doi:10.1016/j.joca.2020.11.006

5.3 A dedicated section on molecular docking is suggested.

Answer: Thank you for your suggestions. A dedicated section on molecular docking was added to the revised manuscript (line 255-269).

  1. The role of others flavonoids on the investigated molecular target is poorly discussed.

Answer: Thank you for your suggestions. The relevant descriptions on the effects of other flavonoids on Nrf2 have been included to the Discussion section (line 241-248).

Round 2

Reviewer 2 Report

Comments and Suggestions for Authors

The authors have adequately addressed the concerns raised by the reviewers and have significantly improved the manuscript in response to the feedback. They provided detailed and thoughtful explanations regarding the potential of Irigenin (IRI) in treating osteoarthritis (OA) and addressed key questions about its therapeutic efficacy, long-term safety, and mechanism of action. Their response demonstrated a solid understanding of the limitations of their current work, including the need for further research into long-term effects and comparison with existing treatments. The authors have also committed to exploring potential drug resistance and interactions with other signaling pathways, which are critical for a comprehensive understanding of IRI's effects. Additionally, they made the necessary grammatical and stylistic revisions to enhance the clarity of the text. Overall, their responses were thorough and aligned with the scientific concerns raised, improving the overall quality and robustness of the study.

Comments on the Quality of English Language

The quality of English in the manuscript is good, the text is well-structured.

Reviewer 3 Report

Comments and Suggestions for Authors

The authors carefully addressed all of my comments, and significant improvements have been made in the current version of the manuscript. I believe the manuscript is now suitable for publication in Pharmaceutics in its current form. It should be noted, however, that the manuscript is not yet formatted according to MDPI's layout.